# Emotion regulation, mindfulness, and self-compassion among patients with borderline personality disorder, compared to healthy control subjects

Ella Salgó[1], Liliána Szeghalmi[1], Bettina Bajzát[1], Eszter Berán[2], Zsolt Unoka[1] *

1 Department of Psychiatry and Psychotherapy, Semmelweis University, Budapest, Hungary, 2 Institute of Psychology, Pázmány Péter Catholic University, Budapest, Hungary

* unoka.zsolt@med.semmelweis-univ.hu

**Data Availability Statement:** All relevant data are within the manuscript and its Supporting information files.

## Abstract

### Objectives

Emotion regulation difficulties are a major characteristic of personality disorders. Our study investigated emotion regulation difficulties that are characteristic of borderline personality disorder (BPD), compared to a healthy control group.

### Methods

Patients with BPD (N = 59) and healthy participants (N = 70) filled out four self-report questionnaires (Cognitive Emotion Regulation Questionnaire, Difficulties in Emotion Regulation Scale, Five Facet Mindfulness Questionnaire, Self-Compassion Scale) that measured the presence or lack of different emotion-regulation strategies. Differences between the BPD and the healthy control group were investigated by Multivariate Analysis of Variance (MANOVA) and univariate post-hoc F-test statistics.

### Results

People suffering from BPD had statistically significantly (p<0.05) higher levels of emotional dysregulation and used more maladaptive emotion-regulation strategies, as well as lower levels of mindfulness and self-compassion compared to the HC group.

### Conclusion

In comparison to a healthy control group, BPD patients show deficits in the following areas: mindfulness, self-compassion and adaptive emotion-regulation strategies. Based on these results, we suggest that teaching emotion-regulation, mindfulness, and self-compassion skills to patients can be crucial in the treatment of borderline personality disorder.

**Funding:** This work was supported by the Hungarian National Research, Development and Innovation Fund [grant numbers NKFI-132546]. PI is Zsolt Unoka.

**Competing interests:** The authors have declared that no competing interests exist.

## 1. Introduction

Emotion regulation consists of the capabilities to process and modulate affective experience. Difficulties with these abilities are often present in people suffering from borderline personality disorder (BPD); moreover, emotion dysregulation is considered a core attribute of this mental disorder [1, 2]. BPD patients are frequently experiencing overwhelming negative emotions such as abandonment, loneliness, jealousy, feeling rejected, hatred, envy, anger, shame and guilt [3–5]. They often report aversive tension, a diffuse, highly aroused state with negative valence [6], and they have difficulties with identifying, naming, or putting into context these emotional states [7–10]. Their reactions to their emotions are often inappropriate: they can be impulsive and have angry outbursts, impulsive behavioral reactions and labile affect. The way they respond to their negative emotions influences the frequency or intensity with which negative affect arises. Their emotion and affect regulation strategies are dysfunctional; for example, they have a tendency towards clinginess [11], dissociation [12], emptiness [13], self-harming behavior [14], alcohol and substance use [15], impulsive sexual behaviors [16], binging, purging [17], and rumination [18]. We hypothesized that they are less able to use functional emotion regulation, such as being mindfully aware of one's emotions, to label, accept and validate emotions, and to tolerate negative or positive emotion-related distress [2].

In the current study, we aimed to investigate whether a broad range of emotion regulation difficulties are characteristic to BPD patients compared to a healthy control group. We also wanted to examine emotion regulation difficulties, adaptive and maladaptive cognitive emotion regulation strategies, mindfulness, and self-compassion in the two groups. Our study is partly a replication and partly an extension of previous studies.

### 1.1 Difficulties in emotion regulation in BPD

Emotion regulation difficulties are a significant characteristic of BPD [1], such that BPD symptoms and interpersonal problems in BPD are found to be mediated by emotion regulation difficulties [19, 20]. The results of a study suggest that emotion dysregulation, particularly lack of access to emotion regulation strategies and lack of emotional clarity, mediate the relationship between BPD symptoms and poor physical health symptoms (e.g., "headaches," "dizziness," "stomach pain") measured eight months later [21]. A study of 100 adults diagnosed with BPD demonstrated significant reductions in emotion dysregulation (measured by DERS) after a six-month-long dialectical behavior therapy intervention [22]. Emotion dysregulation assessed by DERS explained unique variance in BPD symptoms, showing that impulse control difficulties and limited access to emotion regulation strategies have the strongest relationship to BPD [23, 24]. As a consequence of emotion dysregulation, people suffering from BPD show deficits in action planning and emotion regulation functioning as a mechanism of effective and goal-directed behavior [25]. In our study, we would like to compare emotional dysregulation in the BPD and HC groups in an adult sample by using DERS as a measurement tool for emotion dysregulation. The only previous study [26] that compared BPD and HC groups by using DERS analyzed differences in its "acceptance" subscale only. Our study complements these findings by analyzing all subscales of DERS.

### 1.2 Cognitive emotion regulation in BPD measured by CERQ

Cognitive strategies have a crucial role in emotion regulation. In order to measure adaptive and non-adaptive cognitive emotion regulation strategies, the Cognitive Emotion Regulation Questionnaire (CERQ) [27] has been developed, using the following nine subscales: self-blame, other-blame, rumination or focus on thought, catastrophizing, putting into perspective, positive refocusing, positive reappraisal, acceptance and refocus on planning.

Using CERQ, it has been shown that people with BPD tend to practice maladaptive emotion regulation strategies. A study showed [26] that BPD patients have more frequent use of maladaptive cognitive emotion regulation strategies (suppression, rumination, avoidance) and less frequent use of adaptive strategies (acceptance, cognitive reappraisal, problem-solving) relative to HC. Using CERQ, Wijk-Herbrink, and colleagues [28] identified three higher-order factors; adaptive coping, non-adaptive coping, and external attribution style in people with personality disorders. They found that they used more non-adaptive coping and less adaptive coping strategies as compared to a general population sample. This study suggests that dysfunctional cognitive emotion regulation can be a characteristic of personality disorders in general. Another study, however, shows no significant differences between people of cluster B personality disorders and healthy control in the nine cognitive emotion regulation strategies measured by CERQ [29]. Research found [30] that the use of maladaptive cognitive emotion regulation strategies (self-blame, blaming others, rumination, and catastrophizing) were related to high levels of negative affectivity and increased psychological problems in people with PDs. Another study compared BPD and schizotypal PD, where the participants scored similarly on CERQ, except for the catastrophizing subscale that had higher scores in BPD patients [31]. Our study will have added value to the previous studies comparing BPD and HC groups by using CERQ [26, 29, 32], in as much as our research analyzes all the subscales of CERQ and determines effect sizes as well.

## 1.3 BPD and mindfulness

Mindfulness is a non-judgmental, present-focused state of mind characterized by present-moment awareness, where thoughts, perceptions, and feelings are accepted and purposefully brought into attention [33]. Low levels of mindfulness have been proven to play a significant role in personality psychopathology, and specifically in BPD [34]. Mindfulness is inversely associated with BPD features and core areas of dysfunctionality, such as interpersonal ineffectiveness, impulsive, passive emotion regulation, and neuroticism [35, 36]. In a study exploring differences in the five mindfulness facets (measured by FFMQ) among patients diagnosed with either obsessive-compulsive disorder, major depressive disorder or borderline personality disorder and HC, BPD patients scored lower on all mindfulness facets compared to the HC group [37]. In a study conducted by Nicastro et al. [38] fewer mindfulness skills were found in BPD patients than in control participants. Findings demonstrate that dispositional mindfulness is negatively associated with BPD features and suicidal thinking among patients in substance use treatment [39]. The inverse relation between BPD and mindfulness can be explained by the difficulties of BPD patients to be consciously aware of their experiences in the present moment instead of focusing on general concepts. The latter may impair their ability to effectively regulate their emotions [40].

Mindfulness is a multidimensional construct. Yu and Clark [36] investigated the relationship between mindfulness (assessed by FFMQ) and borderline personality traits in a non-clinical sample and found that mindfulness facets relate differentially to BPD symptoms, among them "non-judging" is the facet most strongly related to BPD traits. Research suggests that for BPD patients, mindful self-observation can be an adaptive alternative to rumination when feeling angry [32].

Conceptual integration of mindfulness and emotion regulation was proposed by Chambers, Gullone, and Allen [41]. According to their review, cognitive emotion regulation strategies and mindfulness fundamentally differ in that according to the concept of emotion regulation, unpleasant thoughts/appraisals need to be acted upon or manipulated in some way to make them less distressing. In contrast, mindfulness considers all mental phenomena as mere mental

events that do not need to be transformed. Their proposed "mindful emotion regulation" is the capacity to remain mindfully aware of the experienced emotions, irrespective of their valence, intensity, and without attempting to reappraise or modify them. Based on this proposition, in our study, we consider mindfulness a potential form of emotion regulation. Our study's additional value to the previous research comparing BPD and HC groups by exploring the five mindfulness facets [37] is that it evaluates the effect sizes in terms of the magnitude of the difference between the two groups.

## 1.4 BPD and self-compassion

Self-compassion is a self-regulation strategy that counters self-criticism and related negative self-directed emotions, such as shame [42]. Neff [43] conceptualized self-compassion with the following three dimensions: a) self-kindness vs. self-judgment, b) common humanity vs. isolation, and c) mindfulness vs. over-identification. Based on a quantitative meta-analytic study, each of these factors are suggested to assist adaptive self-regulatory processes [44]. One may reason that such self-regulatory processes in general—including emotion-regulation—are impaired in BPD since BPD is frequently associated with childhood trauma and abuse [45–47], and childhood trauma exposure and emotional dysregulation are suggested to have a complex and bidirectional relationship [48]. Linehan's biosocial theory [49] suggests that what she calls "invalidating environments" during childhood may play an important role in the subsequent development of BPD in adolescence, by hindering the development of self-compassion and emotion-regulation. However, a study [50] found that even though childhood parental invalidation and lack of self-compassion are both strongly associated with BPD symptoms, their associations with BPD symptoms are independent of each other. In contrast, traumatic experiences may contribute to a self-invalidating and self-critical cognitive style [49]. Other studies suggest that self-criticism is a diagnostic element [51] and a frequent characteristic of BPD [52–54].

Research shows that loving-kindness and compassion meditation based on self-compassion lowers self-criticism and improves self-kindness and acceptance in BPD patients [53]. Moreover, self-compassion seems to mediate between mindfulness and BPD symptoms, and between mindfulness and emotion dysregulation as well [55]. Self-compassion is also considered the outcome of mindfulness practice [56].

The above studies suggest that the lack of self-compassion is associated with BPD symptoms and that improved self-compassion can ease the emotional pain experienced in BPD. Some research has already been conducted on comparing BPD population to HC in the context of self-compassion, although with a different aim. Scheibner and colleagues [55] used the Self Compassion Scale (SCS) to compare BPD patients with HC, and found significant differences between these two groups in terms of self-compassion. A study found that BPD patients had significantly higher fears and resistances to all forms of compassion (fears of self-compassion, fears of being open to compassion of others, fears of being compassionate to others) compared to the control group [57]. The current study is an extension of the previous one that compared BPD and HC groups by using SCS [55] since it investigates group differences in the SCS subscales as well.

## 1.5 Mini review of the literature of the studies that compared BPD and HC on one of the following scales: CERQ, DERS, FFMQ, and SCS

Why do we need one further study? As outlined in the Introduction, there are several studies examining emotion regulation difficulties in BPD. However, there are only a few studies

comparing adult BPD groups to healthy control participants, and those that exist do not examine CERQ, DERS, FFMQ and SCS simultaneously by analyzing all of their subscales. We prepared a summary of the literature that compares adult BPD and HC groups by using CERQ, DERS, FFMQ and/or SCS (see Table 1). By administering these four questionnaires in the two groups in the current study, we cover a more comprehensive array of emotion regulation strategies than previous studies.

## 1.6 Hypothesis

We hypothesized that the BPD and HC groups would show significant differences in terms of emotion regulation, mindfulness, and self-compassion. Furthermore, dysfunctional emotion regulation strategies and lack of self-compassion would be predominant among BPD patients. We also hypothesized that adaptive emotion regulation strategies, mindfulness skills, and self-compassion techniques would score higher in the HC group.

## 2. Method

### 2.1 Subjects and procedure

Subjects participated in a four-week-long inpatient psychotherapy program at Semmelweis University's Department of Psychiatry and Psychotherapy between 2017 and 2019. Psychiatrists and clinical psychologists made the diagnoses during intake interviews. Data has been gathered from 59 subjects diagnosed with borderline personality disorder and from 70 healthy control subjects. Medical students recruited age, gender, and education matched healthy control volunteers who were acquaintances and relatives of university students with no known psychiatric disorders. There were 104 female (80.6%) and 25 male (19.4%) participants, with a mean age of 30.7 years (SD = 11.1, range = 18–57). Regarding educational level, 0% completed just the first six years of primary school, 28.7% passed A-level exams, 24.8% did not obtain A-level exams, 3.8% dropped out of college, 9.3% completed vocational studies, 11.6% obtained a college degree, 8.5% dropped out of the university while 13.1% obtained university degree. (To see the distribution of clinical diagnosis, see Table 2).

Subjects had been provided with sufficient information about the research and signed an informed consent sheet. Their anonymity was guaranteed. Participants were diagnosed with SCID II interviews and filled out questionnaires online. The Regional and Institutional Committee of Science and Research Ethics of Semmelweis University approved the research procedure.

### 2.2 Self-reported questionnaires measuring emotion regulation strategies

**The Cognitive Emotion Regulation Questionnaire (CERQ)** is a 36-item questionnaire measuring cognitive emotion regulation strategies applied after having experienced negative life events or situations [27]. It assesses nine cognitive emotion regulation strategies: self-blame, other-blame, rumination, or focus on thought, catastrophizing, putting into perspective, positive refocusing, positive reappraisal, acceptance, and refocus on planning. Cronbach's α coefficients of the subscales in this study ranged between.60 (acceptance) and.89 (positive refocusing). Cognitive emotion regulation strategies were measured on a 5-point Likert scale ranging from 1 (almost never) to 5 (almost always). The Hungarian version of the questionnaire had been validated by Miklósi and colleagues [58].

**The Difficulties in Emotion Regulation Scale (DERS)** [59], was created based on four main aspects of emotion regulation, as defined by the authors:

**Table 1. Mini review of the literature comparing BPD and HC groups based on emotion regulation strategies/dysfunctionalities.**

| Authors, date | Title | Source | Sample | Used scales | Findings | Effect sizes—BPD vs. HC | Subscale analysis |
|---|---|---|---|---|---|---|---|
| Daros et al., 2018 | Cognitive Emotion Regulation Strategies in Borderline Personality Disorder: Diagnostic Comparisons and Associations with Potentially Harmful Behaviors | Psychopathology | BPD (n = 30) MAD (n = 30) HC (n = 32) | **CERQ**, CSI, DASS, **DERS**, MEAQ, RRS, WBSI | BPD subjects endorsed more maladaptive cognitive ER strategies and fewer adaptive strategies compared to HC. Compared to MAD subjects, BPD individuals endorsed more maladaptive cognitive ER strategies, but only when those with subthreshold BPD symptoms in the MAD group were excluded. | **Cohen's d:** CERQ pos refocus: -0.52 CERQ pos reappraisal: -1.52 CERQ putting into persp: -0.56 CERQ refocus on planning: -1.30 CERQ acceptance: -0.05 DERS acceptance (reversed): -2.20 | 5 subscales of CERQ and 1 of DERS were utilized. |
| Didonna et al. 2019 | Relations of mindfulness facets and psychological symptoms among individuals with a diagnosis of Obsessive-Compulsive Disorder, Major Depressive Disorder and Borderline Personality Disorder | Psychology and Psychotherapy Theory Research and Practice | OCD (n = 55), MDD (n = 50), BPD (n = 48), HC (n = 50) | BDI-II, DES, **FFMQ**, SCL-90, TAS-20 | Mindfulness abilities seem to be impaired in psychiatric patients compared with HC. There are disease-specific relationships between some mindfulness facets and specific psychological variables | - | yes |
| Heidari et al., 2015 | Comparative Evaluation of Cognitive Emotion Regulation between "B" Personality Disorders and Normal Persons | Procedia—Social and Behavioral Sciences | AsPD (n = 46) BPD (n = 46) HPD (n = 46) NPD (n = 46) HC (n = 46) | **CERQ**, MCMI III, | There were no significant differences between people of cluster B personality disorders and people of normal personality in nine cognitive coping strategies of CERQ. | **Partial Eta Squared:** CERQ Self-blame: .000 Acceptance: .005 Rumination: .000 Pos refocus: .008 Refocus on planning: .032 Pos reappraisal: .000 Putting into persp: .006 Catastrophizing: .022 Blaming others: .063 | yes |
| Sauer et al., 2016 | Emotion regulation choice in female patients with borderline personality disorder: Findings from self-reports and experimental measures | Psychiatry Research | BPD (n = 24) MD (n = 19) HC (n = 32) | BSL-23, BDI- II, **CERQ**, RSQ-D, SCL-9, | Both patient groups showed maladaptive self-reported emotion regulation choice profiles compared with HC. No differences between the groups in the choice of distraction and reappraisal on the behavioral level and in heart rate responses. In BPD, within-group analyses revealed a positive correlation between symptom severity and the preference for distraction under high-intensity borderline-specific stimuli. | - | yes |
| Scheibner et al., 2017 | Self-Compassion Mediates the Relationship Between Mindfulness and Borderline Personality Disorder Symptoms | Journal of Personality Disorders | BPD (n = 29) HC (n = 30) | FFA, **SCS** | Self-compassion mediates the relationship between mindfulness and BPD symptom severity as well as between mindfulness and emotion dysregulation | **Kappa-squared:** k2 = .20 (Effect size of the indirect effect of mindfulness on BPD symptom severity via self-compassion.) | no |

**Notes**: AsPD = Antisocial Personality Disorder, BPD = Borderline Personality Disorder, BSL-23 = Borderline Symptom List, BDI-II = Beck Depression Inventory, CERQ = Cognitive Emotion Regulation Questionnaire, CSI = Coping Strategies Inventory, DASS = Depression, Anxiety and Stress Scale, DERS = Difficulties in Emotion Regulation Questionnaire, DES = Dissociative Experience Scale, FFA = Freibruger Fragebogen zur Achtsamkeit (Freiburg Mindfulness Inventory), HC = healthy control, HPD = Histrionic Personality Disorder, MAD = mixed anxiety and/or depressive disorder, MCMI III = Millon Clinical Multiaxial Inventory, MD = Major Depression, MEAQ = Multidimensional Experiential Avoidance Questionnaire, NPD = Narcissistic Personality Disorder, RRS = Ruminative Response Style Questionnaire, RSQ-D = Response Style Questionnaire (German version), SCL-9 = Symptoms Checklist 9, SCS = Self-Compassion Scale, TAS-20 = Toronto Alexithymia Scale, WBSI = White Bear Suppression Inventory.

**Table 2. Sociodemographic and clinical variables of patients with borderline personality disorder and healthy comparison subjects.**

| Characteristics | Groups | | | | Test-statistic |
|---|---|---|---|---|---|
| | Borderline Personality disorders (N = 59) | | Healthy Control (N = 70) | | |
| | *Mean* | *SD* | *Mean* | *SD* | *F (x,y)* |
| **Age** | 30.2 | 10 | 31.2 | 12 | 0.2 (1,127) |
| | **N** | **%** | **N** | **%** | χ2 |
| **Gender** | | | | | 1.2 |
| Male | 9 | 15.3 | 16 | 22.9 | |
| Female | 50 | 84.7 | 54 | 77.1 | |
| **Education** | | | | | 9.9 |
| 1. first 6 years of primary school | 0 | 0 | 0 | 0 | |
| 2. A-levels | 18 | 30.5 | 19 | 27.1 | |
| 3. without A-levels | 11 | 18.6 | 21 | 30 | |
| 4. dropped out of college | 3 | 5.1 | 2 | 2.9 | |
| 5. completed vocational studies | 9 | 15.3 | 3 | 4.3 | |
| 6. obtained college degree | 5 | 8.5 | 10 | 14.3 | |
| 7. dropped out of university | 3 | 5.1 | 8 | 11.4 | |
| 8. obtained university degree | 10 | 16.9 | 7 | 10 | |
| **Types of personality disorders** | | | | | |
| Paranoid | 8 | 13.6 | 0 | 0 | |
| Borderline | 59 | 100 | 0 | 0 | |
| Histrionic | 6 | 10.2 | 0 | 0 | |
| Narcissistic | 2 | 3.4 | 0 | 0 | |
| Avoidant | 25 | 42.4 | 0 | 0 | |
| Dependent | 15 | 25.4 | 0 | 0 | |
| Obsessive-compulsive | 14 | 23.7 | 0 | 0 | |
| Passive-Aggressive | 8 | 13.6 | 0 | 0 | |
| Depressive | 26 | 44.1 | 0 | 0 | |
| Schizoid | 1 | 1.7 | 0 | 0 | |
| Schizotypal | 3 | 5.1 | 0 | 0 | |
| **Comorbid disorders** | | | | | |
| Depressive episode | 18 | 40 | 0 | 0 | |
| Generalized Anxiety disorders | 16 | 35.6 | 0 | 0 | |
| Bipolar disorder | 21 | 46.7 | 0 | 0 | |
| Panic disorder | 3 | 6.7 | 0 | 0 | |
| PTSD | 0 | 0 | 0 | 0 | |
| OCD | 1 | 2.2 | 0 | 0 | |
| Psychotic disorder | 4 | 8.9 | 0 | 0 | |
| Substance use disorder | 5 | 11.1 | 0 | 0 | |
| Eating disorder | 7 | 15.6 | 0 | 0 | |
| Somatoform disorder | 1 | 2.2 | 0 | 0 | |

Notes: * p < 0.05. ** p < 0.01.

"(a) awareness and understanding of emotions,

(b) acceptance of emotions,

(c) ability to control impulsive behaviors and behave in accordance with desired goals when experiencing negative emotions,

(d) ability to use situationally appropriate emotion regulation strategies flexibly to modulate emotional responses as desired, in order to meet individual goals and situational demands." (pp42).

Higher scores on the measure indicate greater dysfunctionality or dysregulation. DERS was implemented [59] in its Hungarian version [60] in order to determine the degree of difficulty in emotion regulation. The 36 items of DERS are organized into a 6-factor structure: non-acceptance of emotional responses, difficulty engaging in goal-directed behavior, impulse control difficulties, lack of emotional awareness, limited access to emotion regulation strategies and lack of emotional clarity. Cronbach's α coefficients of the DERS subscales in this research ranged between.67 (impulse control difficulties) and.91 (limited access to emotion regulation strategies). DERS's scales are rated on a 5-point Likert scale.

**The Self-Compassion Scale (SCS)**, developed by Dr. Kristine Neff [43], is applied to measure self-compassion, which is defined as compassion turned inward and refers to how we relate to ourselves in instances of perceived failure, inadequacy or personal suffering [61]. The scale consists of 26 items rated on a 5-point Likert scale. Its three subscales are self-kindness versus self-judgment, a sense of common humanity versus isolation, and mindfulness versus over-identification. Cronbach's α coefficients of the subscales in this study ranged between.56 (self-judgment) and.84 (self-kindness). The Hungarian version of SCS was implemented by Sági and co-workers [62]. In our study, we interpret our findings according to the two-factor model of SCS, which collapses self-kindness, common humanity, and mindfulness items into a positive, "self-compassion" factor and self-judgment, isolation, and over-identification items into a negative, "self-criticism" factor [61].

**The Five Facet Mindfulness Questionnaire** includes 39 items that examine the five major aspects of mindfulness on a 5-point Likert scale: observation, description, mindful actions, non-judgmental inner experience and non-reactivity [63]. Cronbach's α coefficients of the subscales in this study ranged between.70 (non-reactivity) and.88 (description). The Hungarian adaptation of the scale was carried out by Józsa (unpublished work).

## 2.3 Statistical analysis

Our statistical analyses tested the hypothesis that difficulty of emotion regulation scores are higher in patients with borderline personality disorder than in healthy participants against the null-hypothesis of no difference. The differences between the BPD and HC groups in terms of their DERS, CERQ, FFMQ and SCS sub-scales were investigated by Multivariate Analysis of Variance (MANOVA), and subsequently by post-hoc univariate F-test statistics determined from the MANOVA analysis.

The analyses were conducted based on a hierarchical approach. Specifically, first, in our primary analysis, the total score on each of the four scales of interest was tested. Study group (BPD or HC) was used as the independent variable in the MANOVA, whereas DERS-total, CERQ adaptive emotion regulation total, CERQ maladaptive emotion regulation total, FFMS-total, and SCS-total scales served as dependent variables. Second, in case the primary analyses yielded a significant difference, we conducted post-hoc analyses by determining the univariate F-statistics to examine the differences between the two groups in the subscales of the four scales mentioned above. In the post-univariate analyses, we used the Hochberg correction to adjust for the inflation of alpha error as a result of multiple testing. We added an asterisk to those results that remained statistically significant after correction for multiple testing in the tables.

Because of different sample sizes, effect sizes were measured by Hedges' g [64], which provides a measure of effect size weighted according to the relative size of each sample (small

effect = 0.2, medium effect = 0.5, large effect = 0.8, [65]). In order to assess the homogeneity of variances, Levene's test was performed. Where Levene's test indicated unequal variances, a Welch test was performed.

Based on the adopted statistical approach (MANOVA), we conducted a statistical power analysis for our primary comparisons to determine the assay sensitivity (i.e., the statistical effect size for a detectable group difference) in the study The power analysis followed the procedure described in the literature [66, 67]. The input parameters for the computation were the available sample size (n = 59 and 70 in the two groups, respectively), and the required alpha threshold level (= 0.05) and level of correlation in terms of Pearson'r among the individual variables used in the MANOVA analysis. Since the individual measures used in the MANOVA are expected to be correlated for Pearson's we conservatively we adopted a value of 0.5 (i.e., 25% in terms of overlapping variance). Our results indicated that the available sample size provides >80% power to detect a standardized group difference of 0.3 on the variables entered in the MANOVA analysis; this value is considered a small effect size, and was deemed to provide sufficient assay sensitivity for the study.

## 3. Results

### 3.1. Demographic, descriptive and clinical characteristics

The current study included a sample of 129 participants (BPD = 59 (9 males), HC = 70 (16 males)). The two groups did not differ significantly on gender (chi-square test: $\chi 2$ = 1.2, p = 0.27) in levels of education (chi-square test: $\chi 2$ = 9.9, p = 0.12) or in age (ANOVA: (F (1,127) = 0.2; p = 0.62). See Table 2.

### 3.2 MANOVA for the total scores

We conducted MANOVA multivariate statistics to determine whether differences between the means of the BPD and HC groups are statistically significant based on the scales' total scores. The primary MANOVA of the total scores of DERS, CERQ Adaptive, CERQ Non-Adaptive, FFMQ, and SCS found statistically significant differences between the BPD and the HC groups: Multivariate F (5,123) = 61.24, p < .0001; Wilk's $\Lambda$ = 0.29. Results of the post-hoc univariate comparisons are presented in Table 3.

**Table 3. Group comparisons for the total scores of the four scales.**

| Variable | Total Standard Deviation | Pooled Standard Deviation | Between Standard Deviation | R-Square | R-Square / (1-RSq) | F value[a] | Pr > F | g |
|---|---|---|---|---|---|---|---|---|
| DERS Total | 0.7986 | 0.5092 | 0.8691 | 0.5967 | 1.4795 | 187.90 | < .0001 | 2.428 |
| CERQ Adaptive | 0.7179 | 0.5488 | 0.6555 | 0.4201 | 0.7245 | 92.02 | < .0001 | -1.683 |
| CERQ Non-Adaptive | 0.6947 | 0.5469 | 0.6073 | 0.3851 | 0.6263 | 79.54 | < .0001 | 1.589 |
| FFMQ Total | 0.5082 | 0.3619 | 0.5046 | 0.4968 | 0.9874 | 125.40 | < .0001 | -2.002 |
| SCS Total | 0.5040 | 0.2922 | 0.5796 | 0.6665 | 1.9983 | 253.78 | < .0001 | -2.870 |

Notes:

[a]: Univariate post-hoc F-test statistics determined from the MANOVA analysis.

g: Effect size measured by Hedge's g formula.

## 3.3 MANOVA of the two groups based on the difficulty of emotion regulation

Since the primary analyses of DERS total score yielded a significant difference, we conducted post-hoc analyses to examine the differences between the two groups in the subscales of the DERS. In every subscale of DERS, patients with BPD had higher scores than healthy participants (DERS total F(1,127) = 187.90, $p < 0.001$). Effect sizes between the BPD and the HC groups are large, except for one medium effect size in the lack of emotional awareness subscale. Results are presented in Table 4.

Both the primary analyses of "adaptive emotion regulation total" and "maladaptive emotion regulation total" scores yielded a significant difference; we conducted post-hoc analyses to examine the differences between the two groups in the subscales of the CERQ. Only its two subscales, "other-blame" and "acceptance," did not show significant differences between the two groups. Maladaptive emotion regulation strategies scored higher in the BPD group, while adaptive strategies scored higher in the HC group. (CERQ adaptive total F(1,127) = 92.02, p< 0.001, CERQ maladaptive total F(1,127) = 79.54, p< 0.001). Large effect sizes were found between the BPD and HC groups, with the exception of the other-blame and acceptance scales. Negative effect sizes indicate poorer results on the given subscale in the BPD group, e.g., putting into perspective. Results are presented in Table 5.

The FFMQ total score's primary analyses yielded a significant difference (FFMQ total F(1,127) = 125.40, p < 0.001), so we conducted post-hoc analyses to examine the differences between the two groups in its subscales. Four subscales; "mindful actions", "non-judgmental inner experience", "non-reactivity" and "description" had higher scores in the HC group than in the BPD group. Only the "observation" subscale did not present significant differences between the two groups. Effect sizes are medium to large between the two groups, with the exception of the observation subscale that yielded very small effect sizes among the groups. Results are presented in Table 6.

The primary analyses of SCS total score yielded a significant difference, so we conducted post-hoc analyses to examine the differences between the two groups in its subscales. The relevant subscale-pairs in SCS present opposing trends in their mean scores; "self-kindness," "common humanity," and "mindfulness" scored higher in the HC group, while

**Table 4. Group comparisons for the BPD and HC groups on the subscale scores of the difficulties of emotion regulation scale, and effect sizes measured by Hedge's g formula.**

| Measure | Diagnostic groups | | | | Difference among diagnostic groups | | | |
|---|---|---|---|---|---|---|---|---|
| DERS | BPD (N = 59) | | HC (N = 70) | | F[a] | df | p | g |
| | *Mean* | *SD* | *Mean* | *SD* | | | | |
| non-acceptance | 3.04 | 1.02 | 1.91 | 0.81 | 48.69 | 1,127 | < 0.001* | 1.250 |
| difficulty engaging in goal-directed behavior | 3.78 | 0.83 | 2.44 | 0.77 | 90.12 | 1,127 | < 0.001* | 1.679 |
| impulse control difficulties | 3.37 | 0.95 | 1.86 | 0.66 | 111.03[W] | 1,127 | < 0.001* | 1.874 |
| lack of emotional awareness | 3.01 | 0.83 | 2.44 | 0.77 | 16.08 | 1,127 | < 0.001* | 0.714 |
| lack of emotional clarity | 2.82 | 0.95 | 1.79 | 0.70 | 49.32[W] | 1,127 | < 0,001* | 1.250 |
| limited access to emotion regulation strategies | 3.75 | 0.81 | 1.93 | 0.65 | 197.16 | 1,127 | < 0.001* | 2.501 |

Notes:

[a]: Univariate post-hoc F-test statistics determined from the MANOVA analysis.

g: Effect size measured by Hedge's g formula; BPD = patients with borderline personality disorder (1); HC = healthy control (2);

[W]: where Levene's test indicated unequal variances a Welch test was performed. Results that remained statistically significant after correction for multiple testing were marked with an asterisk.

**Table 5. Group comparisons of the BPD and HC groups on the subscale scores of the cognitive emotion regulation questionnaire, and effect sizes measured by Hedge's g formula.**

| Measure | Diagnostic groups | | | | Difference among diagnostic groups | | | |
|---|---|---|---|---|---|---|---|---|
| CERQ | BPD (N = 59) | | HC (N = 70) | | $F^a$ | df | p | g |
| | *Mean* | *SD* | *Mean* | *SD* | | | | |
| self-blame | 3.75 | 0.82 | 2.46 | 0.79 | 81.07 | 1,127 | < 0.001* | 1.594 |
| other-blame | 1.94 | 0.80 | 1.79 | 0.57 | 1.58$^W$ | 1,127 | 0.22 | 0.204 |
| rumination | 3.28 | 0.90 | 2.48 | 0.88 | 25.75 | 1,127 | < 0.001* | 0.899 |
| catastrophizing | 2.95 | 1.05 | 1.74 | 0.73 | 58.38$^W$ | 1,127 | < 0.001* | 1.358 |
| putting into perspective | 2.30 | 0.76 | 2.98 | 0.80 | 24.58 | 1,127 | < 0.001* | -0.869 |
| positive refocusing | 1.64 | 0.58 | 3.08 | 0.92 | 105.62$^W$ | 1,127 | < 0.001* | -1.838 |
| positive reappraisal | 2.20 | 0.81 | 3.77 | 0.85 | 111.82 | 1,127 | < 0.001* | -1.874 |
| acceptance | 2.55 | 0.64 | 2.56 | 0.65 | 0.01 | 1,127 | 0.911 | -0.015 |
| refocus on planning | 2.89 | 1.03 | 3.84 | 0.91 | 30.25 | 1,127 | < 0.001* | -0.972 |

Notes:

[a]: Univariate post-hoc F-test statistics determined from the MANOVA analysis.

g: Effect size measured by Hedge's g formula; BPD = patients with borderline personality disorder (1); HC = healthy control (2);

[W]: where Levene's test indicated unequal variances a Welch test was performed. Results that remained statistically significant after correction for multiple testing were marked with an asterisk.

"self-judgment," "isolation," and "over-identification" have higher scores in the BPD group. (SCS positive subscales total F(1,127) = 82.55, p< 0.001, SCS negative subscales total F(1,127) = 234.00, p< 0.001). Effect sizes are large between the BPD and HC groups. Results are presented in Table 7.

## 4. Discussion

Our study has investigated emotion-regulation, mindfulness, and self-compassion abilities in BPD, compared to HC. Results confirmed our hypothesis that people suffering from BPD had a higher level of emotional dysregulation and used more maladaptive emotion-regulation strategies and less adaptive emotion regulation strategies, lower mindfulness and self-compassion levels than HC participants. We are going to discuss each result in detail below.

**Table 6. Group comparisons of the BPD and HC groups on the subscale scores of the five factor mindfulness questionnaire, and effect sizes measured by Hedge's g formula.**

| Measure | Diagnostic groups | | | | Difference among diagnostic groups | | | |
|---|---|---|---|---|---|---|---|---|
| FFMQ | BPD (N = 59) | | HC (N = 70) | | F | df | p | g |
| | Mean | SD | Mean | SD | | | | |
| observation | 2.93 | 0.79 | 2.94 | 0.71 | 0.00 | 1,127 | 0.962 | -0.013 |
| mindful actions | 2.95 | 0.62 | 3.98 | 0.64 | 82.76 | 1,127 | < 0.001* | -1.616 |
| non-judgmental inner experience | 2.63 | 0.65 | 3.86 | 0.70 | 102.95 | 1,127 | < 0.001* | -1.800 |
| non-reactivity | 2.31 | 0.57 | 2.79 | 0.66 | 19.24 | 1,127 | < 0.001* | -0.789 |
| description | 3.01 | 1.01 | 3.86 | 0.68 | 31.28$^W$ | 1,127 | < 0.001* | -1.003 |

Notes:

a: Univariate post-hoc F-test statistics determined from the MANOVA analysis.

g: Effect size measured by Hedge's g formula; BPD = patients with borderline personality disorder (1); HC = healthy control (2);

[W]: where Levene's test indicated unequal variances a Welch test was performed. Results that remained statistically significant after correction for multiple testing were marked with an asterisk.

**Table 7. Group comparisons of the BPD and HC groups on the subscale scores of the self-compassion scale, and effect sizes measured by Hedge's g formula.**

| Measure | Diagnostic groups | | | | Difference among diagnostic groups | | | |
|---|---|---|---|---|---|---|---|---|
| SCS | BPD (N = 59) | | HC (N = 70) | | F[a] | df | p | g |
| | Mean | SD | Mean | SD | | | | |
| SCS positive subscales total | 2.21 | 0.64 | 3.27 | 0.66 | 82.55 | 1,127 | < 0.001* | -1.58 |
| SCS negative subscales total | 3.95 | 0.53 | 2.25 | 0.69 | 234.00 | 1,127 | < 0.001* | 2.691 |
| self-kindness | 1.89 | 0.78 | 3.24 | 0.88 | 82.32 | 1,127 | < 0.001* | -1.606 |
| self-judgment | 3.89 | 0.69 | 2.34 | 0.86 | 121.12 | 1,127 | < 0.001* | 1.959 |
| common-humanity | 2.12 | 0.73 | 3.19 | 0.93 | 51.00 | 1,127 | < 0.001* | -1.266 |
| isolation | 3.97 | 0.72 | 2.18 | 0.71 | 197.02 | 1,127 | < 0.001* | 2.504 |
| mindfulness | 2.63 | 0.73 | 3.37 | 0.70 | 33.74 | 1,127 | < 0.001* | -1.036 |
| over-identification | 3.98 | 0.60 | 2.23 | 0.74 | 207.22[W] | 1,127 | < 0.001* | 2.574 |

Notes:

[a]: Univariate post-hoc F-test statistics determined from the MANOVA analysis.

g: Effect size measured by Hedge's g formula; BPD = patients with borderline personality disorder (1); HC = healthy control (2);

[W]: where Levene's test indicated unequal variances a Welch test was performed. Results that remained statistically significant after correction for multiple testing were marked with an asterisk.

## 4.1 DERS

In agreement with our hypothesis, results revealed that BPD patients had higher overall emotion dysregulation compared to the HC group. All the six subscales of DERS presented significant differences between the two groups. This result is different from Ibraheim and co-worker's findings in an adolescent sample, where only two subscales ("limited access to strategies" and "impulse control difficulties") differed significantly [24]. The finding is also in agreement with the results of a meta-analysis by Daros and Williams [2]. In this study, results are based on 93 unique studies indicating that symptoms of BPD were associated with less frequent use of adaptive emotion regulation strategies (i.e., problem solving and cognitive reappraisal) and more frequent use of strategies that are less effective in reducing negative affect (i.e. suppression, rumination, and avoidance).

## 4.2 CERQ

Our results show that the BPD and HC populations have significant differences in almost all CERQ subscales-except for "other-blame" and "acceptance". These results are in harmony with a study [68] examining people with BP features after negative mood and rumination induction. Those participants who scored higher on BP features (measured by Morey's Personality Assessment Inventory-Borderline Features Scale [69]) reported higher levels of self-blame. Moreover, self-blame, as well as other-blame seemed to be an indicator of impulsive behavior as well [70]. Social exclusion was also associated with self-blame in BPD patients [71]. Another study shows that self-blame partially mediates the relationship between child maltreatment and later non-suicidal self-injury [72].

Our results demonstrate that the inability to put an unpleasant event into perspective is characteristic of the BPD group. This finding is affirmed by the alternative DSM-5 Model of personality disorders [73] which characterized PDs by impairments in personality functioning and pathological personality traits. The incapability of considering and understanding different perspectives is a defining component of the "empathy" factor of the Levels of Personality Functioning Scale, and a proposed diagnostic criteria for BPD.

### 4.3 Mindfulness

Our findings show impaired mindfulness abilities on four mindfulness facets among BPD patients compared to HC; mindful actions, description, non-reactivity and non-judgmental inner experience. The latter subscale presented the largest difference between the BPD and the HC groups. These results are in agreement with previous studies [42, 74, 75]. The result that the "observing" subscale was not significantly different among the three groups is similar to the finding of Didonna and co-worker's study [37]. Results are in line with the theoretical assumptions that mindfulness practice promotes adaptive emotion regulation strategies [76, 77].

### 4.4 Self-compassion

According to our study, BPD patients scored lower on the adaptive, and higher on the maladaptive dimensions of the self-compassion scale than the healthy control group. Self-compassion has already been examined in BPD in contrast to a healthy population [55, 57]; their findings were similar to our results. A study, where self-compassion was examined in cluster C personality disorders before and after a short-term dynamic psychotherapy, showed that levels of self-compassion increased due to therapy, and this in turn predicted decrease in psychiatric symptoms, and personality pathology [78]. The study of Castilho and co-workers [79] found similar results about self-compassion when examining different clinical samples with diagnoses associated with difficulties in emotion- regulation (e.g. personality disorders).

### 4.5 Limitations

One of the limitations of our study is that self-administered questionnaires might have distorted the data, because self-awareness and self-reflection are impaired functions in BPD [80]. Furthermore, our BPD sample consists of patients participating in a 4 week-long psychotherapy program, suffering from severe symptoms and dysfunctionality; this limits our findings' generalizability to BPD patients who are functioning better or less motivated to seek help. In both of our samples, the number of female participants is much higher than the number of men. This difference reflects a general observation that BPD is diagnosed predominantly (75%) in females in the clinical sample [81], although Grant et al. did not find gender differences in their epidemiologic survey [82]. The differential gender prevalence of BPD in our clinical setting may be the result of clinical sampling bias. In addition, our sample represents BPD patients who seek pharmaco- and psychotherapeutic help, and this is more characteristic to female BPD patients [83].

## 5. Conclusion

In summary, we can conclude that BPD features have a strong association with emotion dysregulation, and that this manifests in emotion regulation strategies—an increased number of maladaptive ones and a decreased number of adaptive ones—as well as in low levels of mindfulness and self-compassion as compared to an HC group. Based on these results, we suggest that teaching emotion-regulation, mindfulness, and self-compassion skills to patients can be crucial in the treatment of borderline personality disorder.

## Supporting information

**S1 File. Dataset to analyze BPD and HC groups based on CERQ, DERS, FFMQ and SCS.**
(XLSX)

## Acknowledgments

We thank Pál Czobor, Ph.D., who is a biostatistician, for his advice on solving statistical questions posed by our reviewers.

## Author Contributions

**Conceptualization:** Zsolt Unoka.

**Data curation:** Ella Salgó, Bettina Bajzát, Zsolt Unoka.

**Formal analysis:** Zsolt Unoka.

**Funding acquisition:** Zsolt Unoka.

**Investigation:** Liliána Szeghalmi, Bettina Bajzát.

**Methodology:** Zsolt Unoka.

**Project administration:** Zsolt Unoka.

**Software:** Bettina Bajzát.

**Supervision:** Zsolt Unoka.

**Writing – original draft:** Ella Salgó, Liliána Szeghalmi, Bettina Bajzát, Eszter Berán, Zsolt Unoka.

**Writing – review & editing:** Ella Salgó, Eszter Berán, Zsolt Unoka.

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
