## [Decision Letter · Decision Letter 0]

6 Aug 2020

PONE-D-20-18458

Emotion regulation, mindfulness, and self-compassion among patients suffering from borderline personality, other personality disorders and healthy control subjects

PLOS ONE

Dear Dr. Unoka,

Thank you for submitting your manuscript to PLOS ONE. After careful consideration, we feel that it has merit but does not fully meet PLOS ONE’s publication criteria as it currently stands. Therefore, we invite you to submit a revised version of the manuscript that addresses the points raised during the review process.

We look forward to receiving your revised manuscript.

Kind regards,

Stephan Doering, M.D.

Academic Editor

PLOS ONE

Reviewers' comments:

Reviewer's Responses to Questions

**Comments to the Author**

1. Is the manuscript technically sound, and do the data support the conclusions?

Reviewer #1: Partly

Reviewer #2: Yes

Reviewer #3: No

2. Has the statistical analysis been performed appropriately and rigorously? 

Reviewer #1: Yes

Reviewer #2: Yes

Reviewer #3: Yes

3. Have the authors made all data underlying the findings in their manuscript fully available?

Reviewer #1: Yes

Reviewer #2: Yes

Reviewer #3: Yes

4. Is the manuscript presented in an intelligible fashion and written in standard English?

Reviewer #1: No

Reviewer #2: Yes

Reviewer #3: No

5. Review Comments to the Author

Reviewer #1: General remarks:

Thois study aims to compare difficulties in emotion regulation in patients with Borderline PD, other PDS and healthy controls. This is an interesting research question and design of the study seems to be appropriate, however sample size seems rather small to answer the research question. I did not find a power analysis calculating the appropriate sample size.

Generally the manuscrit would profit from careful proof-reading and restructuring. The manuscript should follow more strictly general rules for scientific writing (e.g. only results and no interprestations in the results sections). The entire manuscript should be shortened and important aspects should be pointed out more clearly. Connections to other studies should be summarized or also shortened to the most important aspects. References are far too much. Language should be checked carefully. For details see below

Title:

The title only includes “borderline personality”, but the correct term and diagnosis would be “borderline personality disorder”. This should be changed.

Abstract:

Presentation and interpretation of results are mixed up in the passages results and conclusion. This should be presented separately.

Introduction:

The abbreviation BPD is not introduced, please

“Their reactions to their emotions are inappropriate: impulsive and exaggerated expression, angry outbursts, impulsive behavioral reactions and labile affect.” � This statement is to generalized, it should be change to ”…reactions are often/can be/might be...”

There are too many references included in the first passage, you should focus on those which are most important.

Since the introduction describes difficulties in emotion regulation in BPD patients, the research question “are emotional regulation difficulties characteristic for BPD patients” seem trivial. Please give a clearer definition of your research question. Why is this analysis important?

1.1 Difficulties in Emotion Regulation in BPD:

Description of DERS belongs in method section, not in the introduction

Introduction too long, sequence of (too) many studies not linked in an appropriate way. Needs major revision to summarize and clarify.

Methods:

2.1 Subjects and Procedure:

Please provide information how the HC group was recruited?

p. 15 first sentence: “schema modus subscale” seems to be wrong in this sentence, please check. Cronbachs alpha could be summarized (Ranging from xx to xx) to shorten this (and all following) passage.

p. 16 2.2.4 The Five Facet Mindfulness Questionnaire: again “schema modus subsacles”?

2.3 Principal component analysis of the above scales: Please describe more precisely what is the aim of this analysis with different measures.

p. 18 3.2 One-way analysis of variance of the three groups based on difficulty of emotion

regulation: The first sentence belongs to method, not results.

“In each and every subscale of DERS, patients with BPD had the highest scores…” � Why is each and every used? One word would be sufficient for this statement.

3.3 Principal component analysis with promax rotation of the sub-scales of DERS, CREQ, FFMQ and SCS

“…which is well above the acceptable limit of .5 (Kaiser, 1974).” This reference should be numbered like all other references and listed in the reference list. I could not find it there.

p. 29/30 “Effect sizes are medium to high between the PD and HC groups and medium between the BPD and other-PD groups, except for the Blaming Others factor where the effect size between the BPD and other-PD group is zero.”

Where do these effect sizes refer to? The ANOVA reported in Table 7?

3.4. One-way analysis of variance of the three groups based on the four factor of principal component analysis of DERS, CREQ, FFMQ and SCS

In this section only the effect sizes are described. Please describe the main effects of the ANOVA.

In this section results are sometimes related to the hypothesis (p. 21 and 26). Results should be only descriptive without interpretation, so these statements belong to the discussion part of the manuscript.

No correction for multiple testing is mentioned; please describe how you solved the problem of multiple testing and alpha mistake cumulation.

Tables:

The tables are somewhat unclear. Please check carefully if you could design the tables more reader-friendly within the journal’s requirements.

Discussion:

This section should be shorted to most important aspects. This applies to the entire manuscript.

4.1 DERS

“The differing results might be explained by the composition of the samples; Houben and Santangelo used a sample of BPD, Non-BPD and HC groups, however, the Non-BPD group consisted of patients with PTSD, bulimia nervosa, major depressive disorder and panic disorder, while in our research the Non-BPD group involved patients with other PDs.” � Why should patients with axis I diagnoses have more problems with affective dysregulation than patients with axis II diagnoses? Moreover, do you have any information on axis I diagnosis in the other-PD group?

4.2 CERQ

“Using CERQ subscales, results show that both BPD and other-PD groups differ

from HC. Our results also show that the BPD and other-PD populations have significant

differences in almost all CERQ subscales-except for “other-blame” and “acceptance”-

compared to HC.“ � It sounds like the same statement in both sentences, please clarify the difference or leave out one of the sentences.

“These results are in harmony with a study (73) examining people with BP features…” � please specify “BP features”

4.4 Self-compassion

“According to our study, self-compassion seems to be dysfunctional in both BPD and

other-PD groups compared to HCs.” � please check the meaning of this sentence and clarify

“…, and the improvement of this skill can ease the clinical symptoms.” � This conclusion can not be drawn from the present study, because it is only a one-time group.

4.5 Principal Component analysis

“An additional fourth factor, “Other Blame” emerged in our data. This difference might be explained by the fact that Zelkowitz and Cole conducted their research in a non-clinical population, and other-blame is an emotion regulation strategy more characteristic to PDs (72).” � Did the study of Zelkowitz and Cole include a (sub)-scale measuring a similar construct?

Reviewer #2: The manuscript is generally exceptionally clearly written: sentences are short, expressions are unambiguous. In this regard, the manuscript is a pleasure to read. Also, the authors very clearly know their field thoroughly: the reference list is very extensive.

Regarding the Introduction, my only major point is that the second hypothesis comes pretty much “out of the blue”: Zelkowith and Cole have not been mentioned previously and thus it is not clear why the second hypothesis is worthwhile/important to test.

Regarding the Method, it would be helpful to know more about the recruitment and assessment process, e.g.:

- Who made the diagnoses? When (e.g., during intake interviews)? What was the treatment setting (e.g., is this a hospital specializing in treating personality disorders, BPD specifically, etc.)?

- How and from where were the healthy controls recruited?

- Were there any power calculations for the sample size?

Having more information about this would be important for assessing potential selection bias, which is not mentioned at all in the “Limitations”. Further, the “Limitations” is overall very brief.

Some further minor comments below:

p. 5, line 5 – what kind of “poor physical health”?

p. 7 – “interpersonal effectiveness” – perhaps rather “interpersonal ineffectiveness”?

p. 7 – Perhaps what “focus on the present moment” means could be clarified in this context for the reader. Some people might say or think being emotional is “being in the present moment” (e.g., as opposed to rationalizing/intellectualizing). Of course, that’s not the point here, but I think this could be clarified.

p. 8: should it be “lack of self-compassion” rather than “self-compassion” which is associated with BPD?

p.8 “One may reason that such self-regulatory processes in general – including emotion-regulation – are impaired in BPD, since BPD is frequently associated with childhood trauma and/or abuse (54–56).” Please make this link clearer.

p. 38: the last sentence is not clear, please clarify

p. 40 reads “non-mediator” – should be “non-meditator”, I believe

p. 42: “shading light to” probably should be “shedding light on”

There seem to be inconsistencies in Table 2. Sometimes means with two decimals, sometimes with only 1.

Also, the Tables appear quite "rough" visually and definitely not APA (or some such) standard format.

Reviewer #3: Thank you for the oppurtunity to review this manuscript. While I believe that the topic itself is very important and the clinical sample sizes are quite large, there are at least three major issues that need to be addressed before the manuscript can be accepted:

(1) Lack of clarity and motivation for the comparison of BPD and other PD in the introduction.

In the beginning of the introduction, the authors cite many studies that show that emotion regulation difficulties are already well established as a core of BPD. Why do we need one further study? While I understand that it is a very important task to compare BPD not only to a HC group but also other clinical samples, why the authors choose to compare the group to other PD remains unclear. What is the benefit of this comparison? In addition, given that the group of other PD is very heterogenous and due to comorbidities both groups do not even differ on the amount of several diagnosed personality disorders (see table 1), the comparison becomes methodologically questionable. Who are you really comparing against whom here?

At the same time, the introduction is very lengthy and should be shortened to only include the most relevant information.

(2) PCA

The motivation for the PCA remains unclear. In addition, one could question whether it is a good idea to perform a PCA with such a heterogenous sample. The authors should include measures of instability, such as bootstrapping and cross validation.

(3) Language and formatting

Throughout the manuscript there are many language inconsistencies and some formatting mistakes, for example in the tables. As a reviewer, I have only limited time and cannot point out every language mistake, I highly recommend proof reading by a native speaker specialized in research articles.

6. PLOS authors have the option to publish the peer review history of their article (what does this mean?). If published, this will include your full peer review and any attached files.

Reviewer #1: No

Reviewer #2: No

Reviewer #3: No

---

## [Author Response · Author response to Decision Letter 0]

17 Dec 2020

December 16, 2020

Stephan Doering, M.D.

Academic Editor

PLOS ONE

Manuscript Number: PONE-D-20-18458

Title: Emotion regulation, mindfulness, and self-compassion among patients suffering from borderline personality disorder, compared to healthy control subjects

Dear Professor Stephan Doering,

Thank you very much for your letter, which provided us with the opportunity to revise our manuscript.

Based on the helpful suggestions of the reviewers, we have revised the manuscript carefully. I have enclosed a revised version of the above paper for submission to PLOS ONE. We have addressed the comments raised by the reviewers. Following the reviewers' and our biostatistician colleague's advice, we made substantial changes to the manuscript. We omitted the results of the principal component analyses from the manuscript, and we also skipped the patient control group from the analyses. 

Point-by-point responses to the reviewers' comments are listed as follows.

- Editor' requirements on format changes 1:

Response 1:

Based on the first comment of the Editor we made the following changes:

-We used level 1 heading for all major sections and level 2 and level 3 headings for sub-sections.

-We listed corresponding author's initials in parentheses after the email address.

-We used numbers instead of letters to indicate affiliations on the title page.

-We formatted the titles of the tables according to the template.

- Editor' requirements on format changes 2:

Response 2:

Based on the second comment of the Editor, we made the following changes:

-We changed the file name of the supporting file to S1_File.

-We included a Supporting information section at the end of the manuscript:

"The dataset analyzed during the current study is available as a Supporting File (S1_File.xlsx).

S1 File. Dataset to analyze BPD and HC groups based on CERQ, DERS, FFMQ and SCS."

Page 30.

- Reviewer 1 Comment 1/a for the Authors

This is an interesting research question and the design of the study seems to be appropriate, however sample size seems rather small to answer the research question. I did not find a power analysis calculating the appropriate sample size. 

Response 1/a.

We thank the Reviewer for drawing attention to this very important question. Based on the adopted statistical approach (ANOVA), we conducted a statistical power analysis for our primary comparisons to determine the assay sensitivity (i.e., the statistical effect size for a detectable group difference) in the study. The input parameters for the computation were the available sample size (n=59 and 70 in the two groups, respectively), and the required alpha threshold level (=0.01, based on the conservative assumption of independence among the five primary total score measures), and the required statistical power (80%). Our results indicated that the available sample size provides 80% power to detect a standardized group difference of 0.6; this value is considered a medium effect size, and was deemed to be appropriate to detect clinically significant effects in the study. We included this information under section 2.3. on page 15.

Furthermore, following the comments of Reviewer 3, we excluded the other patient group from our analyses; that is why we have only two groups.

- Reviewer 1 Comment 1/b for the Authors

"Generally the manuscript would profit from careful proof-reading" and "Language should be checked carefully."

Response 1/b.

We thank the Reviewer for this comment; we consulted with a native English speaker and corrected our manuscript accordingly.

- Reviewer 1 Comment 2 for the Author

The manuscript should follow more strictly general rules for scientific writing (e.g. only results and no interpretations in the results sections). The entire manuscript should be shortened and important aspects should be pointed out more clearly. 

Response 2.

Thank you for this comment. We reorganized the paper. Now there are only results in the results section. The entire manuscript was shortened, and the important aspects were pointed out more clearly. 

- Reviewer 1 Comment 3 for the Author

Connections to other studies should be summarized or also shortened to the most important aspects. References are far too much. 

Response 3.

Thank you for this remark; we agree with it. Therefore we shortened and summarized the connections to other studies and reduced the number of references.

- Reviewer 1 Comment 4 for the Authors

Title:

The title only includes "borderline personality", but the correct term and diagnosis would be "borderline personality disorder". This should be changed. 

Response 4.

We agree with this suggestion; therefore, we changed the title to 

Emotion regulation, mindfulness, and self-compassion among patients suffering from borderline personality disorder, compared to healthy control subjects.

- Reviewer 1 Comment 5 for the Author

Abstract:

Presentation and interpretation of results are mixed up in the passages results and conclusion. This should be presented separately. 

Response 5.

Thank you for this note. We presented the results and conclusion passages separately in the abstract.

- Reviewer 1 Comment 6 for the Author

Introduction:

The abbreviation BPD is not introduced, please. 

Response 6.

Thank you for this note. We introduced the abbreviation of BPD in the abstract and in the introduction as well. 

Page 2 and 3.

- Reviewer 1 Comment 7 for the Author

"Their reactions to their emotions are inappropriate: impulsive and exaggerated expressions, angry outbursts, impulsive behavioral reactions and labile affect." This statement is too generalized, it should be changed to"…reactions are often/can be/might be..." 

Response 7.

Thank you for this note. We changed the sentence to the following: 

"Their reactions to their emotions are often inappropriate: they can be impulsive and have angry outbursts, impulsive behavioral reactions, and labile affect."

Page 3.

- Reviewer 1 Comment 8 for the Author

There are too many references included in the first passage, you should focus on those which are most important. 

Response 8.

Thank you for this note. We reduced the number of references to the most important ones.

 - Reviewer 1 Comment 9 for the Author

Since the introduction describes difficulties in emotion regulation in BPD patients, the research question "are emotional regulation difficulties characteristic for BPD patients" seem trivial. Please give a clearer definition of your research question. Why is this analysis important?

Response 9.

Thank you for this note. We gave a more precise definition of our research question:

"In the current study, our aim was to investigate whether certain emotion regulation difficulties are specifically characteristic to BPD patients, compared to a healthy control group. We also wanted to examine difficulties in emotion regulation, adaptive and maladaptive cognitive emotion regulation strategies, mindfulness and self-compassion in the two groups." 

Page 4.

- Reviewer 1 Comment 10 for the Author

1.1 Difficulties in Emotion Regulation in BPD:

Description of DERS belongs in the method section, not in the introduction.

Response 10.

Thank you for this note. We transposed the description of DERS in the method section.

Page 12.

- Reviewer 1 Comment 11 for the Author

Introduction too long, sequence of (too) many studies not linked in an appropriate way. Needs major revision to summarize and clarify. 

Response 11.

Thank you for this note. We revised the introduction.

- Reviewer 1 Comment 12 for the Author

Methods:

2.1 Subjects and Procedure:

Please provide information how the HC group was recruited? 

Response 12.

Thank you for this note. We provided information on the recruitment of HC group members.

"Healthy control volunteers were recruited by medical students, and they were acquaintances and relatives of university students with no known psychiatric disorders." 

Page 9.

- Reviewer 1 Comment 13 for the Author

p. 15 first sentence: "schema modus subscale" seems to be wrong in this sentence, please check. 

Response 13.

Thank you for this note. We modified this sentence.

- Reviewer 1 Comment 14 for the Author

Cronbach's alpha could be summarized (Ranging from xx to xx) to shorten this (and all following) passage. 

Response 14.

Thank you for this note. We summarized Cronbach's alphas according to your advice in this and all the following passages.

Page 12, 13 and 14.

- Reviewer 1 Comment 15 for the Author

p. 16 2.2.4 The Five Facet Mindfulness Questionnaire: again "schema modus subscales"?

Response 15.

Thank you for this note. We corrected this sentence.

- Reviewer 1 Comment 16 for the Author

2.3 Principal component analysis of the above scales: Please describe more precisely what is the aim of this analysis with different measures. 

Response 16.

Thank you for this note. Based on the comments of Reviewer 3 and the advice of a biostatistician, we excluded the PCA from our paper. 

- Reviewer 1 Comment 17 for the Author

p. 18 3.2 One-way analysis of variance of the three groups based on difficulty of emotion

regulation: The first sentence belongs to method, not results. 

Response 17.

Thank you for this note. We transposed this first sentence to the method section (2.4).

- Reviewer 1 Comment 18 for the Author

"In each and every subscale of DERS, patients with BPD had the highest scores…" Why is each and every used? One word would be sufficient for this statement. 

Response 18.

Thank you for this note. We deleted 'each' from this sentence.

Page 15.

- Reviewer 1 Comment 19 for the Author

3.3 Principal component analysis with promax rotation of the subscales of DERS, CREQ, FFMQ and SCS

"…which is well above the acceptable limit of .5 (Kaiser, 1974)." This reference should be numbered like all other references and listed in the reference list. I could not find it there. 

Response 19.

Thank you for this note. We left the PCA out of the analysis (please see Response 16).

- Reviewer 1 Comment 20 for the Author

p. 29/30 "Effect sizes are medium to high between the PD and HC groups and medium between the BPD and other-PD groups, except for the Blaming Others factor where the effect size between the BPD and other-PD group is zero."

Where do these effect sizes refer to? The ANOVA reported in Table 7? 

Response 20.

Thank you for this note. This sentence referred to the ANOVA reported in Table 7, but since it is repeated in section 3.4, we took the sentence out.

 - Reviewer 1 Comment 21 for the Author

3.4. One-way analysis of variance of the three groups based on the four factor of principal component analysis of DERS, CREQ, FFMQ and SCS

In this section only the effect sizes are described. Please describe the main effects of the ANOVA. 

Response 21.

Thank you for this note. We decided to leave the PCA out of our analysis. (Please see Response 16 to Reviewer 1.)

- Reviewer 1 Comment 22 for the Author

In this section results are sometimes related to the hypothesis (p. 21 and 26). Results should be only descriptive without interpretation, so these statements belong to the discussion part of the manuscript. 

Response 22.

Thank you for this note. We deleted the interpretative sentences from the results section.

- Reviewer 1 Comment 23 for the Author 

No correction for multiple testing is mentioned; please describe how you solved the problem of multiple testing and alpha mistake cumulation. 

Response 23.

Thank you for this note. Specifically, first, in our primary analysis, the total score on each of the four scales of interest was tested using the Hochberg correction to adjust for the inflation of alpha error as a result of multiple testing. Study group (BPD or HC) was used as the independent variable in the ANOVA, whereas DERS-total, CERQ adaptive emotion regulation total, CERQ maladaptive emotion regulation total, FFMS-total, and SCS-total scales served as dependent variables. Second, in case the primary analyses yielded a significant difference, we conducted post-hoc analyses to examine the differences between the two groups in the subscales of the four scales mentioned above. Analogous to the primary comparisons, we also applied Hochberg correction in these analyses in order to adjust for the alpha error inflation. In the tables we added an asterisk to those results that remained statistically significant after correction for multiple testing.

Page 14.

- Reviewer 1 Comment 24 for the Author

Tables:

The tables are somewhat unclear. Please check carefully if you could design the tables more reader-friendly within the journal's requirements.

Response 24.

Thank you for this note. We redesigned the tables.

- Reviewer 1 Comment 25 for the Author

Discussion:

This section should be shortened to the most important aspects. This applies to the entire manuscript.

4.1 DERS

"The differing results might be explained by the composition of the samples; Houben and Santangelo used a sample of BPD, Non-BPD and HC groups, however, the Non-BPD group consisted of patients with PTSD, bulimia nervosa, major depressive disorder and panic disorder, while in our research the Non-BPD group involved patients with other PDs." Why should patients with axis I diagnoses have more problems with affective dysregulation than patients with axis II diagnoses? Moreover, do you have any information on axis I diagnosis in the other-PD group? 

Response 25.

Thank you for this note. Due to the major changes we made in our analysis we left this sentence out entirely. The axis I diagnoses are included in Table 1 titled "Sociodemographic and clinical variables of patients with borderline personality disorder and healthy comparison subjects".

Page 10-11.

- Reviewer 1 Comment 26 for the Author

4.2 CERQ

"Using CERQ subscales, results show that both BPD and other-PD groups differ

from HC. Our results also show that the BPD and other-PD populations have significant

differences in almost all CERQ subscales-except for "other-blame" and "acceptance"-

compared to HC. "It sounds like the same statement in both sentences, please clarify the difference or leave out one of the sentences.

Response 26.

Thank you for this note. We rewrote this sentence and left the redundant parts out.

Page 21.

- Reviewer 1 Comment 27 for the Author

"These results are in harmony with a study (73) examining people with BP features…" please specify "BP features" 

Response 27.

Thank you for this note. BP features had been assessed by Morey's Personality Assessment Inventory (BAI-POR), we added this information in the text and the references section.

Page 21.

- Reviewer 1 Comment 28 for the Author

4.4 Self-compassion

"According to our study, self-compassion seems to be dysfunctional in both BPD and

other-PD groups compared to HCs." please check the meaning of this sentence and clarify

Response 28.

Thank you for this note. We left out this sentence.

- Reviewer 1 Comment 29 for the Author

"…, and the improvement of this skill can ease the clinical symptoms." This conclusion can not be drawn from the present study, because it is only a one-time group. 

Response 29.

Thank you for this note. We left out this sentence.

- Reviewer 1 Comment 30 for the Author

4.5 Principal Component analysis

"An additional fourth factor, "Other Blame" emerged in our data. This difference might be explained by the fact that Zelkowitz and Cole conducted their research in a non-clinical population, and other-blame is an emotion regulation strategy more characteristic to PDs (72)." à Did the study of Zelkowitz and Cole include a (sub)-scale measuring a similar construct?

Response 30.

Thank you for this note. We decided to leave out the PCA from our article, please see Response 16. 

- Reviewer 2 Comment 1 for the Author

Regarding the Introduction, my only major point is that the second hypothesis comes pretty much "out of the blue": Zelkowith and Cole have not been mentioned previously and thus it is not clear why the second hypothesis is worthwhile/important to test.

Response 1.

Thank you for this note. We left out the PCA from our article, please see Response 16 to Reviewer 1.

- Reviewer 2 Comment 2 for the Author

Regarding the Method, it would be helpful to know more about the recruitment and assessment process, e.g.:

- Who made the diagnoses? When (e.g., during intake interviews)? What was the treatment setting (e.g., is this a hospital specializing in treating personality disorders, BPD specifically, etc.)?

- How and from where were the healthy controls recruited? 

Response 2.

Thank you for your comments. The diagnoses were made by psychiatrists and clinical psychologists during intake interviews. As for the treatment setting, this is a hospital providing inpatient treatment (4 week-long psychotherapy programs) for people suffering from personality disorders. We added this information in the Method section.

We added information about the healthy controls: 

"The age, gender, and education matched healthy control volunteers were recruited by medical students and they were acquaintances and relatives of university students with no known psychiatric disorders." 

Page 9.

- Reviewer 2 Comment 3 for the Author

- Were there any power calculations for the sample size? 

Response 3.

Thank you for your comment. Please see Response 1 to Reviewer '.

Page 15

- Reviewer 2 Comment 4 for the Author

Having more information about this would be important for assessing potential selection bias, which is not mentioned at all in the "Limitations". Further, the "Limitations" is overall very brief.

Response 4.

Thank you for your comment. We extended the "Limitations" section by adding the following:

"One of the limitations of our study is that self-administered questionnaires might have distorted the data. Moreover, our BPD sample consists of patients participating in a 4 week-long psychotherapy program, suffering from severe symptoms and dysfunctionality; this limits our findings' generalizability to BPD patients who are functioning better, or who are less motivated to change. In both of our samples the number of female participants is much higher than the number of men, but this difference reflects a general observation that BPD is diagnosed predominantly (%75) in females (77). Whether the differential gender prevalence of BPD in clinical settings is the result of sampling bias is still a question."

Page 24

- Reviewer 2 Comment 5 for the Author

Some further minor comments below:

p. 5, line 5 – what kind of "poor physical health"?

Response 5.

Thank you for your comment. In this quoted research the Cohen-Hoberman Inventory of Physical Symptoms—Revised (CHIPS-R; Campbell, Greeson, Bybee, & Raja, 2008) was used to assess physical health symptoms. Participants were presented with a list of 35 commonly experienced physical symptoms (e.g., "headaches," "dizziness," "stomach pain") and asked to indicate how much each physical health problem had bothered or distressed them during the past four months (including the current day) on a scale from 0 (not at all) to 4 (extreme bother). Items are summed to create an overall index of physical health symptoms. We changed the commented sentence to the following:

"The results of a study suggest that emotion dysregulation, particularly lack of access to emotion regulation strategies and lack of emotional clarity mediate the relationship between BPD symptoms and poor physical health symptoms (e.g., "headaches," "dizziness," "stomach pain") measured 8 months later ."

Page 4.

- Reviewer 2 Comment 6 for the Author

p. 7 – "interpersonal effectiveness" – perhaps rather "interpersonal ineffectiveness"?

Response 6.

Thank you for your comment. We corrected the phrase to interpersonal ineffectiveness.

Page 6

- Reviewer 2 Comment 7 for the Author

p. 7 – Perhaps what "focus on the present moment" means could be clarified in this context for the reader. Some people might say or think being emotional is "being in the present moment" (e.g., as opposed to rationalizing/intellectualizing). Of course, that's not the point here, but I think this could be clarified.

Response 7.

Thank you for your comment. We corrected this sentence to "The inverse relation between BPD and mindfulness can be explained by the difficulties of BPD patients to be consciously aware of their experiences in the present moment instead of focusing on general concepts. The latter may impair their ability to effectively regulate their emotions (26). "

Page 6.

- Reviewer 2 Comment 8 for the Author

p. 8: should it be "lack of self-compassion" rather than "self-compassion" which is associated with BPD? 

Response 8.

Thank you for the comment. We corrected the phrase to "lack of self-compassion".

Page 7

- Reviewer 2 Comment 9 for the Author

p.8 "One may reason that such self-regulatory processes in general – including emotion-regulation – are impaired in BPD, since BPD is frequently associated with childhood trauma and/or abuse (54–56)." Please make this link clearer.

Response 9.

Thank you for your comment. We complemented the sentence:

"One may reason that such self-regulatory processes in general – including emotion-regulation – are impaired in BPD since BPD is frequently associated with childhood trauma and abuse (49–51), and childhood trauma exposure and emotional dysregulation are suggested to have a complex and bidirectional relationship (78)."

Page 7.

- Reviewer 2 Comment 10 for the Author

p. 38: the last sentence is not clear, please clarify

Response 10.

Thank you for your comment. We modified the last sentence to "Based on these results, we suggest that teaching emotion-regulation, mindfulness and self-compassion skills to patients can be crucial in the treatment of borderline personality disorder." 

Page 23.

- Reviewer 2 Comment 11 for the Author

p. 40 reads "non-mediator" – should be "non-meditator", I believe

Response 11.

Thank you for your comment. We corrected the typo. This part of the original paper had been omitted.

- Reviewer 2 Comment 12 for the Author

p. 42: "shading light to" probably should be "shedding light on"

Response 12.

Thank you for your comment. This part of the original paper had been omitted.

- Reviewer 2 Comment 13 for the Author

There seem to be inconsistencies in Table 2. Sometimes means with two decimals, sometimes with only 1.

Also, the Tables appear quite "rough" visually and definitely not APA (or some such)standard format.

Response 13.

Thank you for your comments. We reformatted the tables and corrected the inconsistencies.

- Reviewer 3 Comment 1/a for the Author

(1) Lack of clarity and motivation for the comparison of BPD and other PD in the introduction.

In the beginning of the introduction, the authors cite many studies that show that emotion regulation difficulties are already well established as a core of BPD. Why do we need one further study? While I understand that it is a very important task to compare BPD not only to a HC group but also other clinical samples, why the authors choose to compare the group to other PD remains unclear. What is the benefit of this comparison? In addition, given that the group of other PD is very heterogenous and due to comorbidities both groups do not even differ on the amount of several diagnosed personality disorders (see table 1), the comparison becomes methodologically questionable. Who are you really comparing against whom here?

Response 1/a.

Thank you for your comment. Based on your comment, we discussed these problems with a biostatistician, and following his advice, we left out the other-PD group from the comparison and we also omitted the PCA.

- Reviewer 3 Comment 1/b for the Author

At the same time, the introduction is very lengthy and should be shortened to only include the most relevant information.

Response 1/b.

Thank you for this note. We reorganized the Introduction. (see Response 2 and 3 for Reviewer 1.).

- Reviewer 3 Comment 2/a for the Author

(2) PCA

The motivation for the PCA remains unclear. In addition, one could question whether it is a good idea to perform a PCA with such a heterogenous sample. The authors should include measures of instability, such as bootstrapping and cross validation.

Response 2/a.

Thank you for this note. We discussed these problems with a biostatistician, and he advised us to leave out the PCA from our article.

- Reviewer 3 Comment 2/b for the Author

(3) Language and formatting

Throughout the manuscript there are many language inconsistencies and some formatting mistakes, for example in the tables. As a reviewer, I have only limited time and cannot point out every language mistake, I highly recommend proofreading by a native speaker specialized in research articles.

Response 2/b.

Thank you for your comment. We consulted with a native English speaker and corrected our manuscript accordingly (see Response 1/b for Reviewer 1).

---

## [Decision Letter · Decision Letter 1]

18 Jan 2021

PONE-D-20-18458R1

Emotion regulation, mindfulness, and self-compassion among patients who have a borderline personality disorder, compared to healthy control subjects

PLOS ONE

Dear Dr. Unoka,

Thank you for submitting your manuscript to PLOS ONE. After careful consideration, we feel that it has merit but does not fully meet PLOS ONE’s publication criteria as it currently stands. Therefore, we invite you to submit a revised version of the manuscript that addresses the points raised during the review process.

We look forward to receiving your revised manuscript.

Kind regards,

Stephan Doering, M.D.

Academic Editor

PLOS ONE

Reviewers' comments:

Reviewer's Responses to Questions

**Comments to the Author**

1. If the authors have adequately addressed your comments raised in a previous round of review and you feel that this manuscript is now acceptable for publication, you may indicate that here to bypass the “Comments to the Author” section, enter your conflict of interest statement in the “Confidential to Editor” section, and submit your "Accept" recommendation.

Reviewer #2: All comments have been addressed

Reviewer #3: (No Response)

2. Is the manuscript technically sound, and do the data support the conclusions?

Reviewer #2: (No Response)

Reviewer #3: Yes

3. Has the statistical analysis been performed appropriately and rigorously? 

Reviewer #2: (No Response)

Reviewer #3: No

4. Have the authors made all data underlying the findings in their manuscript fully available?

Reviewer #2: (No Response)

Reviewer #3: Yes

5. Is the manuscript presented in an intelligible fashion and written in standard English?

Reviewer #2: (No Response)

Reviewer #3: Yes

6. Review Comments to the Author

Reviewer #2: (No Response)

Reviewer #3: Major issue:

You responded to my critique about the other PD group and the PCA by simply omitting both of them. I find this problematic, because it does not answer the question “why do we need one further study?” And leaves you with a very basic design.

In particular, at the end of your introduction, please rewrite the following paragraph:

“We hypothesized that they are less able to use functional emotion regulation, such as being mindfully aware of one's emotions, to label, accept and validate emotions, and to tolerate negative or positive emotion-related distress without non-adaptive reactivity, putting into perspective, positive refocusing, positive reappraisal of the situation and refocus on planning. In the current study, we aimed to investigate whether certain emotion regulation difficulties are specifically characteristic of BPD patients compared to a healthy control group. We also wanted to examine emotion regulation difficulties, adaptive and maladaptive cognitive emotion regulation strategies, mindfulness, and self-compassion in the two groups.”

And please explain how your study is different from previous studies or how, if it is not different, it is still valuable to have additional data.

Minor issues:

“Our study investigated emotion regulation difficulties that are characteristic of Borderline Personality Disorder (BPD), compared to a healthy control group “

Borderline personality disorder should not be capitalized

“In comparison to a healthy control group, BPD patients have a serious problem in the following areas:”

“have a serious problem” is colloquial and cannot be proven from the data.

“In our study, we would like to compare emotional dysregulation in the BPD and HC groups in an adult sample by using DERS as a measurement tool for emotion dysregulation.”

Either explain how this is different from previous studies or at least recognize:

“In our study, we would like to replicate previous findings and …”

“between people of cluster B PDs” – omit the blank space between B and P.

“We also assumed that adaptive emotion regulation strategies, mindfulness skills, and self-compassion techniques would score higher in the HC group.”

“We also hypothesized that…”

Differences among the BPD and HC groups in terms of their DERS, CERQ, FFMQ and SCS sub-scales were investigated by One-way Analysis of Variance (ANOVA).

If you have only two groups, you do not need an ANOVA to test for differences but a simple t-test would suffice. However, if you want to test for group differences on more than one scale, it could make sense to calculate a multivariate analysis of variance (MANOVA) with all 4 scales or their subscales as dependent measures, instead of correcting for multiple testing. It should not make major differences in the results, however I would advise with your statistician.

“One of the limitations of our study is that self-administered questionnaires might have distorted the data.”

Please elaborate.

7. PLOS authors have the option to publish the peer review history of their article (what does this mean?). If published, this will include your full peer review and any attached files.

Reviewer #2: No

Reviewer #3: No

---

## [Author Response · Author response to Decision Letter 1]

23 Feb 2021

February 23, 2021

Stephan Doering, M.D.

Academic Editor

PLOS ONE

Manuscript Number: PONE-D-20-18458

Title: Emotion regulation, mindfulness, and self-compassion among patients with borderline personality disorder, compared to healthy control subjects

Dear Professor Stephan Doering,

Thank you very much for your letter, which provided us with the opportunity to revise our manuscript. 

Based on the helpful suggestions of the reviewers, we have revised the manuscript carefully. I have enclosed a revised version of the above paper for submission to PLOS ONE. We have addressed the comments raised by the reviewers.

Point-by-point responses to the reviewers’ comments are listed as follows. 

- Reviewer 3 Comment 1

You responded to my critique about the other PD group and the PCA by simply omitting both of them. I find this problematic, because it does not answer the question “why do we need one further study?” And leaves you with a very basic design.

In particular, at the end of your introduction, please rewrite the following paragraph:

“We hypothesized that they are less able to use functional emotion regulation, such as being mindfully aware of one's emotions, to label, accept and validate emotions, and to tolerate negative or positive emotion-related distress without non-adaptive reactivity, putting into perspective, positive refocusing, positive reappraisal of the situation and refocus on planning. In the current study, we aimed to investigate whether certain emotion regulation difficulties are specifically characteristic of BPD patients compared to a healthy control group. We also wanted to examine emotion regulation difficulties, adaptive and maladaptive cognitive emotion regulation strategies, mindfulness, and self-compassion in the two groups.”

And please explain how your study is different from previous studies or how, if it is not different, it is still valuable to have additional data.

Response 1

We thank the Reviewer for drawing attention to this very important question. We added a new subsection under 1.6 (Mini-review of the literature) where we explain how or study is different from previous ones:

“Why do we need one further study? As outlined in the Introduction, there are several studies examining emotion regulation difficulties in BPD. However, there are only a few studies comparing adult BPD groups to healthy control participants, and those that exist do not examine CERQ, DERS, FFMQ and SCS simultaneously by analyzing all of their subscales. We prepared a summary of the literature that compares adult BPD and HC groups by using CERQ, DERS, FFMQ and/or SCS (see Table 1). By administering these four questionnaires in the two groups in the current study, we cover a more comprehensive array of emotion regulation strategies than previous studies.”

We also modified the above quoted paragraph:

“In the current study, we aimed to investigate whether a broad range of emotion regulation difficulties are characteristic to BPD patients compared to a healthy control group. We also wanted to examine emotion regulation difficulties, adaptive and maladaptive cognitive emotion regulation strategies, mindfulness, and self-compassion in the two groups. Our study is partly a replication and partly an extension of previous studies.”

- Reviewer 3 Comment 2

“Our study investigated emotion regulation difficulties that are characteristic of Borderline Personality Disorder (BPD), compared to a healthy control group “

Borderline personality disorder should not be capitalized

Response 2.

We thank the Reviewer for this comment. We corrected this mistake. 

- Reviewer 3 Comment 3 

“In comparison to a healthy control group, BPD patients have a serious problem in the following areas:”

“have a serious problem” is colloquial and cannot be proven from the data.

Response 3.

Thank you for this comment. We rephrased this sentence to the following:

“In comparison to a healthy control group, BPD patients show deficits in the following areas: mindfulness, self-compassion and adaptive emotion-regulation strategies.”

- Reviewer 3 Comment 4 

“In our study, we would like to compare emotional dysregulation in the BPD and HC groups in an adult sample by using DERS as a measurement tool for emotion dysregulation.”

Either explain how this is different from previous studies or at least recognize:

“In our study, we would like to replicate previous findings and …”

Response 4.

Thank you for this comment. Please see Response 1. 

- Reviewer 3 Comment 5 

“between people of cluster B PDs” – omit the blank space between B and P.

Response 5.

Thank you for your comment. We corrected this part to “between people of cluster B personality disorders”.

- Reviewer 3 Comment 6 

“We also assumed that adaptive emotion regulation strategies, mindfulness skills, and self-compassion techniques would score higher in the HC group.”

“We also hypothesized that…” 

Response 6.

Thank you for this comment. We rewrote the sentence accordingly.

- Reviewer 3 Comment 7 

Differences among the BPD and HC groups in terms of their DERS, CERQ, FFMQ and SCS sub-scales were investigated by One-way Analysis of Variance (ANOVA).

If you have only two groups, you do not need an ANOVA to test for differences but a simple t-test would suffice. However, if you want to test for group differences on more than one scale, it could make sense to calculate a multivariate analysis of variance (MANOVA) with all 4 scales or their subscales as dependent measures, instead of correcting for multiple testing. It should not make major differences in the results, however I would advise with your statistician.

Response 7.

Thank you for this comment. We calculated a multivariate analysis of variance. 

Page 8.

“Our statistical analyses tested the hypothesis that difficulty of emotion regulation scores are higher in patients with borderline personality disorder than in healthy participants against the null-hypothesis of no difference. The differences between the BPD and HC groups in terms of their DERS, CERQ, FFMQ and SCS sub-scales were investigated by Multivariate Analysis of Variance (MANOVA), and subsequently by post-hoc univariate F-test statistics determined from the MANOVA analysis. 

The analyses were conducted based on a hierarchical approach. Specifically, first, in our primary analysis, the total score on each of the four scales of interest was tested. Study group (BPD or HC) was used as the independent variable in the MANOVA, whereas DERS-total, CERQ adaptive emotion regulation total, CERQ maladaptive emotion regulation total, FFMS-total, and SCS-total scales served as dependent variables. Second, in case the primary analyses yielded a significant difference, we conducted post-hoc analyses by determining the univariate F-statistics to examine the differences between the two groups in the subscales of the four scales mentioned above.”

- Reviewer 3 Comment 8

“One of the limitations of our study is that self-administered questionnaires might have distorted the data.”

Please elaborate.

Response 8.

Thank you for this comment. We modified this sentence to the following: 

“One of the limitations of our study is that self-administered questionnaires might have distorted the data, because self-awareness and self-reflection are impaired functions in BPD (83).”

---

## [Editor Report · Decision Letter 2]

26 Feb 2021

Emotion regulation, mindfulness, and self-compassion among patients with borderline personality disorder, compared to healthy control subjects

PONE-D-20-18458R2

Dear Dr. Unoka,

We’re pleased to inform you that your manuscript has been judged scientifically suitable for publication and will be formally accepted for publication once it meets all outstanding technical requirements.

Kind regards,

Stephan Doering, M.D.

Academic Editor

PLOS ONE

---

## [Editor Report · Acceptance letter]

3 Mar 2021

PONE-D-20-18458R2 

Emotion regulation, mindfulness, and self-compassion among patients with borderline personality disorder, compared to healthy control subjects 

Dear Dr. Unoka:

I'm pleased to inform you that your manuscript has been deemed suitable for publication in PLOS ONE. Congratulations! Your manuscript is now with our production department. 

Kind regards, 

on behalf of

Professor Stephan Doering 

Academic Editor

PLOS ONE